# Catalytic Sabatier Process under Thermally and Magnetically Induced Heating: A Comparative Case Study for Titania-Supported Nickel Catalyst

**DOI:** 10.3390/nano13091474

**Published:** 2023-04-26

**Authors:** Sourav Ghosh, Sharad Gupta, Manon Gregoire, Thibault Ourlin, Pier-Francesco Fazzini, Edmond Abi-Aad, Christophe Poupin, Bruno Chaudret

**Affiliations:** 1Laboratoire de Physique et Chimie des Nano-Objets (LPCNO), Université de Toulouse, CNRS, INSA, UPS, 31077 Toulouse, France; thibault.ourlin@gmail.com (T.O.); pierfrancesco.fazzini@cemes.fr (P.-F.F.); chaudret@insa-toulouse.fr (B.C.); 2Unité de Chimie Environnementale et Interactions sur le Vivant (UCEIV), UR 4492, Université du Littoral Côte d’Opale, 145 Avenue Maurice Schumann, 59140 Dunkerque, France; manon.gregoire@univ-littoral.fr (M.G.); edmond.abiaad@univ-littoral.fr (E.A.-A.); christophe.poupin@univ-littoral.fr (C.P.)

**Keywords:** CO_2_ methanation, supported nickel catalyst, induction heating, catalyst stability

## Abstract

In the present paper, we compare the activity, selectivity, and stability of a supported nickel catalyst in classical heating conditions and in magnetically activated catalysis by using iron wool as a heating agent. The catalyst, 5 wt% Ni supported on titania (Degussa P25), was prepared via an organometallic decomposition method and was thoroughly characterized by using elemental, microscopic, and diffraction techniques. In the event of magnetic induction heating, the % CO_2_ conversion reached a maximum of ~85% compared to ~78% for thermal conditions at a slightly lower temperature (~335 °C) than the thermal heating (380 °C). More importantly, both processes were found to be stable for 45 h on stream. Moreover, the effects of magnetic induction and classical heating over the catalyst evolution were discussed. This study demonstrated the potential of magnetic heating-mediated methanation, which is currently under investigation for the development of pilot-scale reactors.

## 1. Introduction

The current dependence on fossil-based products ranging from fuels to consumable goods in our daily life has caused a continuous upsurge of the atmospheric carbon dioxide (CO_2_) concentration, which has a mitigating effect on the global environment [1]. The catalytic conversion of CO_2_ to chemicals and fuels represents an attractive approach for the valorization of CO_2_ and carbon neutrality [2,3,4,5,6]. The central idea behind the catalytic conversion of CO_2_ to feedstocks is the utilization of CO_2_ produced in chemical or biological processes and for chemically storing the energy produced by renewable energies, whether as electricity or, after electrolysis, as hydrogen. Among the different options for transforming CO_2_, its hydrogenation into methane seems to be particularly promising since methane can be easily injected into the existing gas delivery pipes [7,8,9,10,11,12]. This process was termed a power-to-gas (PtG) process and has recently gained a substantial amount of attention from the scientific community [11,12,13,14].

The CO_2_ methanation process is strongly exothermic (${\mathrm{CO}}_{2}\;\left(g\right)+4H_{2}\;\left(g\right)\to {\;\mathrm{CH}}_{4}\;\left(g\right)+2H_{2}O\;\left(g\right)$; $\Delta H_{298}=-165.0\;\mathrm{kJ}\cdot {\mathrm{mol}}^{-1}$, $\Delta G_{298}=-113.2\;\mathrm{kJ}\cdot {\mathrm{mol}}^{-1}$) [8]. However, the activation of CO_2_ faces significant kinetic limits and requires the use of efficient catalysts. Upon the discovery of the CO_2_ methanation process, over the last century, significant progress has been achieved in terms of catalyst development [2,3,4,5,6,7,8,9,10]. Among the most active metals identified and tested for CO_2_ methanation reaction, the most popular ones are based on nickel, which represents the best compromise between cost and activity [8,9,10,15,16,17]. The catalytic methanation performance of a catalyst depends on many factors, such as metal loading, promoter addition, supports, catalyst preparation, catalytic reaction condition, and the choice of catalytic reactors [8,9,10,15,16,17].

Industrially, CO_2_ methanation is typically carried out in continuous flow mode by using fixed-bed reactors under conventional heating conditions. However, the intrinsic exothermicity of the CO_2_ methanation reaction (catalytic hot spots) in the large-scale reactor often renders the process complicated in terms of additional heat management and possible security issues [18]. Furthermore, the catalytic hot spots might have a long-term effect on the catalytic performances and the respective stability. Apart from modifying the reactors in terms of heat management for the highly exothermic processes, an alternate technology was recently envisioned in terms of effective heat management and the subsequent reactor temperature [19]. 

In the case of magnetically induced heating, ferromagnetic/electrically conductive materials release heat through hysteresis and eddy losses, when placed in a high-frequency alternating magnetic field (AMF) [19]. Generally, magnetic heating permits the system to reach a high temperature within a few seconds due to the high heating ramp. Hence, this system is suitable for energy intermittence, a main feature of renewable sources. Another advantage associated with the magnetically induced heating process is contactless heating (direct heating of the magnetic/electrically conductive materials through adsorption/conversion of electromagnetic energy). Hence, by using magnetic materials, the heat is rapidly and homogeneously disseminated within the catalytic bed without the need for heating the whole reactor system [19,20]. The reaction is assumed to be limited only by kinetics and not by heat transfer, as shown for the efficient steam reforming of methane [21], CO_2_ hydrogenation [22], and water electrolysis [23]. Over the past few years, we have developed the magnetic-induction-mediated, heterogeneous catalytic process, which spun over from gas phase CO_2_ methanation [24,25,26,27,28,29], propane dehydrogenation and propane reforming [30,31], Fischer–Tropsch synthesis [32], and hydrodeoxygenation in the liquid phase [33,34].

Since the first proofs of concept, magnetic-heating-mediated CO_2_ methanation, for which we have recently demonstrated both the energy efficiency and the possibility to develop a process [25], has been extensively investigated. We have described a new methodology that makes use of commercial iron (Fe) wool as a heating agent, along with supported nickel (Ni) catalysts. This system was found to be highly active, methane-selective, dynamically responsive towards intermittency, and energy-efficient [25]. However, one crucial question remains about the long-term stability of such a system compared to catalysts operated in conventional set-ups. One point of attention is the stress induced by the very fast heating of the catalyst compared to conventional processes. The second point is the possible catalyst degradation due to the presence of the heating elements in their close vicinity. To address these questions, we have undertaken a comparison of the catalytic activity and long-term stability of a similar catalytic system based on the best formulation we have obtained so far (mixture of Fe wool and Ni/TiO_2_ catalyst, [25]) in both conventional and magnetically induced conditions. The study shows that the activities are comparable, slightly better for the magnetically induced system, and that there is no specific loss of activity when using magnetic induction-mediated catalysis.

## 2. Experimental Section

### 2.1. Materials

Bis(1,5-cyclooctadiene)nickel(0) (Ni(COD)_2_, 98%) and titania (P25 TiO_2_, >98%) were purchased from Strem (Newburyport, MA, USA) and Acros Organics (Geel, Belgium), respectively. The titania support was dried in a hot air oven (~100 °C) and subsequently transferred to the glove box for storage. Toluene (>99%) and mesitylene (99%) were purchased from Fisher Scientific (Loughborough, UK) and VWR (Radnor, PA, USA), respectively. The solvents were dried in a solvent purifier and bubbled with Argon for 20 min before they were transferred into a glove box. Silicon carbide (SiC, coarse powder, 46 grit) was obtained from Fisher Scientific. Iron wool (Paille de fer FINE, commercial name) was obtained from Gerlon (Abbeville, France). Basic characterization details of the commercial Fe wool have been reported elsewhere [25].

### 2.2. Synthesis of Ni/TiO_2_ Catalyst

A titania-supported Ni nanoparticle catalyst was synthesized by following the previous report from our group [25]. The targeted nickel loading was 5 wt%. In a typical synthesis, 2 g of titania and 0.493 g of Ni(COD)_2_ were placed in a Fischer–Porter (FP) bottle, and 30 mL of mesitylene was added. The yellow color reaction mixture was vigorously stirred for 1 h at room temperature. Subsequently, the FP bottle was transferred into a preheated oil bath at 150 °C and stirred for 1 h. After the completion of the reaction, the dark grey solid was collected through decantation, and the solid was washed with 10 mL of toluene four times. Subsequently, the solid was dried for 6 h under a dynamic vacuum. The powder was stored inside an argon-filled glove box. The sample was abbreviated as 5Ni/TiO_2_.

### 2.3. Characterization

The powder X-ray diffraction (XRD) pattern was acquired by using a PANalytical EMPYREAN diffractometer () equipped with a cobalt source (Kα—1.79 Å). The measurement conditions were as follows: 35 kV voltage, 45 mA current, step size—0.0263°, counting time—197 s, total duration—50 min. The magnetic property measurements were carried out by using a vibrating scanning magnetometer (PPMS Evercool II from Quantum Design, San Diego, CA, USA). The magnetization vs magnetic field (M-H) data (hysteresis loop) were obtained at two different temperatures (300 and 5 K) up to ±3 T external field. The magnetic susceptibility concerning the temperature (5–300 K range) at field-cooled (FC) and zero-field-cooled (ZFC) conditions was recorded at a 50 Oe field. Scanning electron microscopy (SEM) images were obtained by using a JEOL JSM-7800F microscope (Akishima, Tokyo, Japan). The SEM-EDS elemental mappings were acquired by using an EDS Bruker XFlash detector. The Inductively Coupled Plasma—Atomic Emission Spectroscopy (ICP-AES) method was used for the determination of nickel content within the sample by using an iCAP 6300 ICP Duo Spectrometer (Thermo Fisher Scientific, Waltham, MA, USA). Bright-field transmission electron microscopy (BFTEM) imaging was carried out by using a JEOL JEM 1400 TEM microscope (Akishima, Tokyo, Japan) (operating voltage 120 kV). ImageJ software was used for the calculation of the nickel particles size. The high-resolution transmission electron microscopy (HRTEM) imaging was carried out by using a JEOL JEM-ARM200F microscope (Akishima, Tokyo, Japan) equipped with a probe Cs corrector and cold FEG gun. The high-angle annular dark field (HAADF) scanning transmission electron microscopy (STEM) imaging and the respective energy-dispersive X-ray spectroscopy (EDS) elemental mapping were accomplished by using a high-angle SDD CENTURIO-X detector attached to the microscope.

### 2.4. Catalytic Test

Under thermal heating, a catalytic CO_2_ methanation reaction was carried out inside a tubular fixed-bed, 8 mm diameter stainless steel reactor that was heated by a vertical electric furnace and equipped with a temperature program controller. The temperature was controlled by a K-type thermocouple, which was placed inside the reactor and on top of the catalytic bed. The gas composition at the inlet to the reactor was CO_2_/H_2_/N_2_ = 10/40/50, with a total flow rate of 100 mL·min^−1^ (GHSV: 20,000 mL·g_cat_^−1^·h^−1^). The outlet gases of the reactor (CO_2_, CO, CH_4_, and H_2_) were analyzed with an online micro-chromatograph (Global Analyzer Solutions) that was equipped with two thermal conductivity detectors (TCD). The CO_2_ gas was separated by using the Porapak T column, while the H_2_, CO, and CH_4_ gases are separated by using the Molsieve 5A column. These compounds are detected by using thermal conductivity detectors (TCD_1_ and TCD_2_). The tests were performed in the temperature range from 120 to 400 °C, with the sample kept at steady-state operation for 30 min at each temperature. The heating rate between steps was 5 °C min^−1^. The stability test was carried out at 380 °C for 45 h, under similar experimental conditions in terms of the gas mixture. Equilibrium CO_2_ conversion was calculated by using the HSC Chemistry 5.0 software. The quantification of the reagents (CO_2_ and H_2_) and products (CO and CH_4_) present in the reaction was performed from the values obtained from the chromatograms and the calibrations carried out for each gas. The general formulas used for the calculations of CO_2_ conversion, CH_4_ yield, and selectivity are given below.
(1)$$X_{CO_{2}}=\left(\frac{A_{CH_{4}}\times RF_{CH_{4}}+A_{CO}\times RF_{CO}}{A_{CO_{2}}\times RF_{CO_{2}}+A_{CH_{4}}\times RF_{CH_{4}}+A_{CO}\times RF_{CO}}\right)\times 100$$
(2)$$S_{CH_{4}}=\left(\frac{A_{CH_{4}}\times RF_{CH_{4}}}{A_{CH_{4}}\times RF_{CH_{4}}+A_{CO}\times RF_{CO}}\right)\times 100$$
(3)$$Y_{CH_{4}}=\left(S_{CH_{4}}\times X_{CO_{2}}\right)\times 100$$

Here, $A_{X}$ corresponds to the area of component *X* in a GC chromatogram, and $RF_{X}$ corresponds to the response factor of component *X* obtained from a GC calibration curve.

A CO_2_ methanation reaction under magnetic heating conditions was accomplished inside a borosilicate glass reactor (internal diameter = 1 cm). The catalyst was placed in the middle of the reactor over a porous silica bed support. The reactor was placed vertically inside a coil capable of generating an alternating magnetic field (AMF) with 100 kHz frequency (manufactured by ID partner, Grenoble, France). In each experiment, 0.3 g of catalyst was loaded, along with 0.3 g of commercial iron wool, and the K-type thermocouple was placed at the top of the catalyst bed with the help of a glass capillary. The gas composition for the inlet was maintained as CO_2_:H_2_:Ar = 10:40:50. The total flow was maintained at 100 mL·min^−1^ (gas hourly space velocity: 20,000 mL·g_cat_^−1^·h^−1^). The experiments were carried out in the temperature range from 120 to 400 °C, and at each temperature, the sample was kept for 30 min under steady-state conditions. The temperature was modulated by varying the amplitude of the applied magnetic field (0–12 mT). In the case of H_2_ pre-treatment, the sample was held at approximately ~400 °C for 4 h under pure H_2_ flow (30 mL·min^−1^). The stability test was carried out at 360 °C for 45 h, under the same reaction condition in terms of the gas mixture. For security reasons, the reactive gas mixture was switched to the inert argon gas flow without turning off the coil (and subsequent heating of the catalytic bed) overnight. The outlet gas mixture was analyzed by using gas chromatography (PerkinElmer Clarus 580, Waltham, MA, USA) coupled with a mass spectrometer (PerkinElmer Clarus SQ8T, Waltham, MA, USA) and a thermal conductivity detector (TCD). The CO_2_ conversion, methane yield, and methane selectivity were calculated by using Equations (1)–(3). The RF factor of each compound for the TCD detector was calculated by injecting a known quantity of these compounds.

## 3. Results and Discussion

### 3.1. Catalyst Characterization

Titania (P25 TiO_2_)-supported nickel catalyst (5 wt%) was synthesized by using an organometallic decomposition method in the presence of a titania support [25]. The reported synthesis was quite robust and scalable up to gram scale. Powder X-ray diffraction pattern of the as-prepared 5Ni/TiO_2_ is shown on Figure 1a. The diffraction peaks can be assigned to the two different crystallographic phases of titania, namely, rutile (ICDD: 00-021-1276) and anatase (ICDD: 00-004-0850). In addition, a tiny broad peak resembling the (111) lattice plane of fcc (face-centered cubic) phase of nickel was detected. The weak nature of the nickel diffraction peak can be ascribed to the poor crystalline nature of Ni nanoparticles. The magnetic measurement study and TEM imaging study further validate the state of the Ni nanoparticle (further discussed in the following section). Moreover, the low loading of Ni is another possible reason behind the weak diffraction signal. Elemental analysis for nickel was carried out by using the ICP-AES technique. The nickel loading for the as-prepared sample was found to be 5%.

The magnetic properties of the as-prepared 5Ni/TiO_2_ sample are presented in Figure 1. Figure 1b shows the magnetization vs field curve measured at 300 K and 5 K. The magnetization values were normalized concerning the total Ni content within the sample. The sample shows room-temperature superparamagnetic behavior, whereas, at 5 K, the Ni nanoparticles display ferromagnetic behavior. At 5 K, the M-H curve measured at a high field (3 T) does not become saturated. This observation further points to the possible contribution of paramagnetic species present in the sample. The paramagnetic contribution was removed (Appendix A). The saturation magnetization value at 300 K was 9.4 emu·g_Ni_^−1^. The spontaneous magnetization of 5Ni/TiO_2_ (18.3 emu·g_Ni_^−1^) measured at 5 K was slightly higher than the one reported in the literature for 10Ni/TiO_2_ (16.4 emu·g_Ni_^−1^). This minute difference could be explained in terms of their average particle size; the bigger Ni particles show higher spontaneous magnetization than the smaller particles [25]. The saturation magnetization value reported here is significantly lower than the bulk nickel (55 emu·g_Ni_^−1^) [35]. This decrease could be attributable to the poor crystallinity of nickel nanoparticles. Upon cooling down at 5 K under an applied magnetic field of 3 T, no exchange bias was observed, indicating the absence of surface oxidation of nickel. At 5 K, the coercive field was 24 mT. The presence of blocking temperature (T_B_) on the ZFC curve at 118 K is the signature of a ferromagnetic-to-paramagnetic transition. From a qualitative analysis of the shape of the FC curve on the FC-ZFC plot, it could be concluded that the interaction between the magnetic Ni nanoparticles is negligible.

Figure 1d shows the BFTEM image. The nickel nanoparticles were well dispersed on the titania support; the average particle size calculated from the histogram was 6.7 ± 1.7 nm (Appendix A). The HRTEM image and the respective fast Fourier transform (FFT) pattern at the reciprocal space are shown in Figure 1f,g, respectively. The HRTEM image of 5Ni/TiO_2_ shows the lattice fringes, and the corresponding FFT pattern could be indexed to the fcc phase of bulk Ni (<110> zone axis orientation). The STEM HAADF image and the STEM-EDS elemental mapping are shown in Figure 2. The STEM HAADF image shows that the nickel nanoparticles are homogeneously distributed on the titania support. This observation was further validated by using the STEM-EDS elemental mapping. The SEM image and the corresponding EDS elemental mapping are shown in Appendix A. The SEM-EDS elemental mapping also demonstrates an even distribution of Ni over the titania support. The Ni concentration from the EDS spectra was measured to be 4.1 wt%. All these characterization data prove that the 5Ni/TiO_2_ sample is composed of titania homogeneously decorated with Ni nanoparticles.

### 3.2. Catalytic Performances

Thermal catalytic CO_2_ methanation was studied for a 5Ni/TiO_2_ catalyst under different reaction conditions, which are shown in Table 1. Figure 3a shows the catalytic performances of the 5Ni/TiO_2_ catalyst in the CO_2_ conversion by using different parameters. In all cases, the catalyst showed negligible CO_2_ conversion at temperatures below 200 °C. Conversion of CO_2_ increases with temperature. Although the methanation reaction is exothermic, the activation of the CO_2_ molecule, which is very stable, requires a significant energy input due to its chemical inertia. This energy is provided in thermal form. The GHSV has a minor effect on early conversion at the low reaction temperature. It was observed that the catalytic activity increases after pre-treating the catalyst with H_2_. This can be related to the complete reduction of Ni under H_2_. The 5Ni/TiO_2_ catalyst with SiC additives and H_2_ pre-treated conditions gave the best CO_2_ conversion value (~78%) at 380 °C. Apart from the complete reduction of Ni under H_2_, the presence of SiC probably inhibits the agglomeration of Ni nanoparticles by controlling the exothermicity of the process.

Figure 3b shows the CH_4_ and CO selectivity of the catalyst at different reaction temperatures. At the initial temperature window (150–300 °C), a slight effect on CH_4_ formation and loss of selectivity due to the formation of some CO was noted for the 5Ni/TiO_2_ (150 mg) catalyst because of the higher GHSV (40,000 mL·g_cat_^−1^·h^−1^). However, at a higher reaction temperature > 300 °C, the CH_4_ selectivity was >99%. Nevertheless, except for 5Ni/TiO_2_ (150 mg), all catalytic conditions lead to >99% selectivity for CH_4_ at all reaction temperatures. Appendix A displays the CH_4_ yield concerning the reaction temperature. It clearly shows that the GHSV (40,000 mL·g_cat_^−1^·h^−1^—black trace vs. 20,000 mL·g_cat_^−1^·h^−1^—red trace) has a minor effect on the rate of CH_4_ formation at a lower reaction temperature. The H_2_ pre-treatment increases the overall methane formation, whereas dilution of the catalytic bed with SiC has a minor positive effect on methane formation.

In the case of the magnetically induced heating process, the catalytic bed is typically composed of a catalyst (5Ni/TiO_2_) and a heating agent (Fe wool). The heating of the catalytic composite bed is usually carried out by applying an alternating magnetic field (AMF) inside a coil operating with a frequency of 100 kHz [25,26]. The heating response upon turning on the magnetic field is instantaneous. The rate of heating is high (70–100 °C·min^−1^). The beneficial role of Fe wool as a heating agent under the influence of a magnetic field was discussed in our previous work [25]. In a control experiment, under similar experimental conditions, pure Fe wool did not show any catalytic activity (Appendix A). At first, the as-prepared 5Ni/TiO_2_ catalyst (300 mg) was tested as a model system for comparing the catalytic performances between classical and magnetic-induction-mediated processes. Furthermore, after studying the catalytic reactions under classical heating conditions, the importance of the SiC addition and the H_2_ pre-treatment step was identified. Hence, a similar composition was tested under magnetically induced heating. Figure 4 shows the % CO_2_ conversion and % CH_4_ selectivity plots against the temperature for the reactions carried out under the magnetically induced heating condition. In a control experiment, pure Fe wool under the same experimental condition does not display any catalytic activity. As expected, the SiC diluted and H_2_ pre-treated catalyst showed a higher % CO_2_ conversion than the pure 5Ni/TiO_2_ catalyst without any diluent and pre-treatment (Figure 4a). For both composite beds, >99% CH_4_ selectivity at all reaction temperatures was observed (Figure 4b).

A comparative catalytic activity analysis was performed between thermal and magnetically induced heating-mediated reactions (Table 2). Figure 5 shows the % CO_2_ conversion plot concerning the reaction temperature for 300 mg of 5Ni/TiO_2_ (black trace) and 300 mg 5Ni/TiO_2_ + SiC + H_2_ pre-treated (red trace) catalysts under both thermal and magnetic induction heating. In the case of magnetic induction heating and for catalytic test conditions similar to those used in thermal catalysis, a catalytic activity at a lower temperature than that observed for thermal catalysis was observed, probably due to the local hot spot generated by magnetic materials under alternating magnetic field [19,22]. Unlike the thermally heated reactors, in the case of magnetic heating, it is difficult to measure the actual catalyst surface temperature by using a thermocouple. Hence, the average temperature noted at the top of the catalytic bed is slightly underestimated compared to the actual temperature of the core of the catalytic bed [23,32,37,38,39]. The temperature difference (ΔT) between thermal and magnetic induction heating was found to be ~30–40 °C to reach the same % CO_2_ conversion (~20% conversion, within the kinetic regime) (Figure 5). To date, the measurements of catalyst surface temperature remain a daunting task [40]. Furthermore, with the low Ni loading and high heating capabilities of iron wool, the actual surface temperature of the Ni nanoparticles is perhaps slightly underestimated. The temperature differences noted here for 5Ni/TiO_2_ are in the range of temperature differences reported for 10Ni/SiC systems in the literature by Truong-Phuoc et al. [39].

In the literature, Pham-Huu and their co-workers have thoroughly investigated the catalytic CO_2_ methanation performances (% CO_2_ conversion, % CH_4_ selectivity versus temperature) under both induction heating and classical heating conditions [22,38,39,41]. It was reported that, at a given temperature, the catalytic % CO_2_ conversion and %CH_4_ selectivity values were higher for induction heating conditions than for classical heating. The catalysts investigated were composed of Ni nanoparticles that were supported on non-conductive Al_2_O_3_ [38] and on thermally and electrically conductive carbon supports [22,41]. In the case of non-conductive Al_2_O_3_ as a support, a higher concentration of Ni was loaded, which has been exploited as a heating cum catalytic agent [38]. In the case of a SiC supported Ni catalyst, SiC acts as a thermal conductor for better heat management, and the role of Ni loading was investigated in a scenario in which additional graphite felt was utilized as a heating agent [39]. In the case of electrically conductive carbon supports, both oxidized carbon felt disc [22] and activated carbon [41] were utilized as heating agents under a magnetic field.

In previous work, we demonstrated that the Fe wool as an alternating heating agent working inside the air-cooled 100 kHz coil with low power consumption can be a judicious choice for enhancing the experimental energy efficiency [25,26]. The energy efficiency of a process is the ratio between the output calorific values and the total energy input, and it is an indicator for implementing the technology for the PtG process [42]. The use of macroscopic Fe wool heated under AMF via eddy current not only enhances the energy efficiency of a process but also opens up the avenue for choosing a highly active, supported nickel catalyst with lower Ni loading. Herein, the productivity (${\mathrm{mol}}_{CH_{4}}\cdot g_{\mathrm{Ni}}^{-1}\cdot h^{-1}$) of the 5Ni/TiO_2_ catalyst tested under different thermal and magnetic induction heating conditions was calculated and tabulated in Appendix A. The productivity of the 5Ni/TiO_2_ catalyst working under magnetically induced heating conditions was found to be better than that of the thermally heated process (Appendix A). Similar differences in their activity between the thermal and magnetic induction heating-mediated processes have been observed by Pham-Huu and their co-workers [22,38,39,41]. The productivity noted here is higher than the one reported in the literature for the supported Ni system working under magnetic induction heating and similar operational condition [38,39,41]. The origin of the higher productivity is the combination of the higher GHSV and the low Ni loading.

All spent catalysts were investigated by using TEM and XRD techniques. The particle size calculated from the bright field TEM images is listed in Appendix A. In general, the average particle size increases slightly after the catalytic test. In both cases, under classical and magnetic induction heating conditions, the average particle size of the Ni nanoparticles after the catalytic test remains almost similar. The powder XRD stack plot of the spent catalysts and the as-prepared 5Ni/TiO_2_ sample are shown in Appendix A. The diffraction peaks of Ni nanoparticles of thermally catalyzed samples are broad and low in intensity; hence, the crystallite size was not calculated. In the case of magnetically induced catalysis, the possible contamination of post-catalytic samples with the Fe wool renders the crystallite size calculation difficult [25].

The effect of magnetic induction heating over classical heating would be most prominent over magnetic elements, which is Ni. To elucidate the effect of thermal and magnetic heating over Ni nanoparticles, the characterization of the magnetic properties has been carried out for a spent 5Ni/TiO_2_ catalyst. A spent (300 mg) 5Ni/TiO_2_ catalyst was chosen as a model system to rule out the differences originating from the possible SiC contamination in other samples. The saturation magnetization values obtained at 300 and 5 K for the induction-heated system were significantly higher than the classically heated system (Appendix A). This observation indicates that the Ni nanoparticles treated under induction heating crystallize at a higher extent than those treated under classical heating. Under the influence of the magnetic field, the Ni nanoparticles may locally heat and act as a local catalytic hotspot for the methanation reaction. On the other hand, Ni nanoparticles catalyze the methanation reaction by using only the external heat provided within the system under thermal conditions. The possible overheating of the local Ni sites under magnetic induction could be responsible for its higher catalytic activity at a given reaction temperature and its possible agglomeration. The nature of the ZFC and FC curves for spent catalysts obtained after the thermal and the magnetic induction-mediated reactions is comparable even if an increase in the presence of possible dipolar interactions between magnetic Ni nanoparticles in both conditions can be detected (Appendix A). Overall, the crystallinity and the particle size of the Ni nanoparticles supported on TiO_2_ after the catalytic test are only slightly modified concerning the as-prepared samples.

A stability test of 45 h was carried out at 380 °C for the H_2_ pre-treated SiC diluted 5Ni/TiO_2_ 300 mg catalyst mix. As shown in Figure 6a, the %CH_4_ selectivity was relatively stable (>99%) for the 5Ni/TiO_2_ catalyst, indicating appropriate catalytic stability under thermal heating conditions. Here, the H_2_ pre-treatment induces the formation of stable Ni nanoparticles, leading to a stable activity during the initial time on stream. Moreover, SiC prevents the hot spot formation in the catalytic bed and prevents the agglomeration of Ni nanoparticles during the exothermic CO_2_ methanation reaction. The Ni nanoparticle size obtained from the TEM analysis of the spent catalyst is 7.1 ± 1.3 nm, which is only slightly larger than the initial nanoparticles (6.7 ± 1.7 nm) (Appendix A). The yield of CO remained at <1% during the stability test. Similarly, a stability test of 45 h was conducted at 360 °C for the same catalytic bed composition and identical gas flow conditions under magnetically induced heating conditions (Figure 6b). The % CO_2_ conversion and % CH_4_ selectivity were stable throughout the time on stream. The TEM analysis of the post-catalytic test samples again indicates a minor increase in the Ni nanoparticle size (7.4 ± 1.5 nm) as compared to the fresh catalyst (6.7 ± 1.7 nm) (Appendix A). The BF TEM micrographs of the two spent catalysts after the stability test are shown in Appendix A. The powder XRD diffraction pattern of the spent catalyst after the stability test under thermal and magnetically induced heating conditions is shown in Appendix A. In terms of stability, the 5Ni/TiO_2_ catalyst performs well under both magnetic and classical heating conditions. Moreover, the local heating of the magnetic Ni nanoparticles and close contact with the Fe wool (heating agent) under the magnetic field does not affect its long-term catalytic performances. It could be concluded from this study that the Fe wool and 5Ni/TiO_2_ composite bed is a stable and promising system for long-term methanation reaction under magnetic induction heating conditions.

## 4. Conclusions

In this work, a simple gram scale synthesis of 5Ni/TiO_2_ was presented. Using different microscopic and spectroscopic techniques, the structure of the catalyst was elucidated as being homogeneously distributed Ni nanoparticles over the TiO_2_ support. The catalytic experiments involving 5Ni/TiO_2_ were carried out both in thermal and magnetic induction heating conditions. In order to compare the catalytic performances of the 5Ni/TiO_2_, a thermal catalytic test was first carried out under various reaction conditions, and the potential catalytic test condition was identified. The catalytic activity of the 5Ni/TiO_2_ at similar reaction conditions (150–400 °C; H_2_/CO_2_/Ar = 10/40/50, total flow = 100 mL·min^−1^; GHSV = 20,000 mL·g_cat_^−1^·h^−1^) under magnetic heating conditions showed higher % CO_2_ conversion with >99% CH_4_ selectivity at a lower reaction temperature than the classical heating mediated process. The slight advantages in terms of catalytic activities of the magnetic induction heating over classical heating could be attributed to the possible generation of local hotspots (close to the catalytically active sites) as well as better heat management under induction heating. The long-term stability test of a 5Ni/TiO_2_ catalyst over ~45 h on stream under both thermal and magnetic heating conditions was carried out. The % CO_2_ conversion and % CH_4_ selectivity were found to be stable under different heating conditions. This investigation pointed out the potential of the Fe wool and of the active methanation catalyst composite system, which could be a long-term solution for its implementation towards the development of a pilot scale reactor working under induction heating conditions, which is currently under investigation.

## Figures and Tables

**Figure 1 nanomaterials-13-01474-f001:**
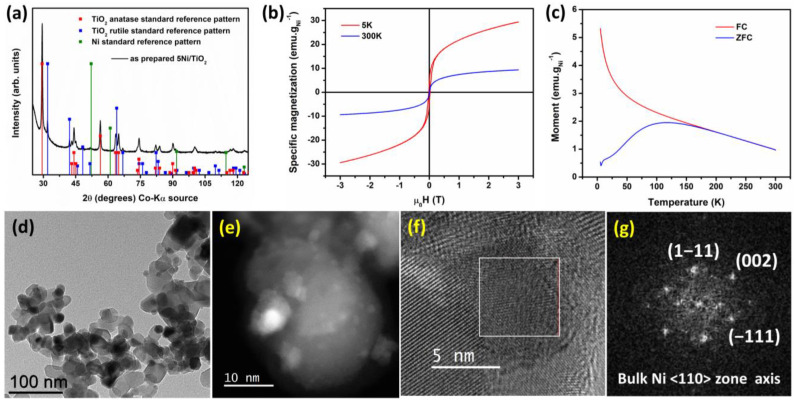
Characterization data for sample 5Ni/TiO_2_: (**a**) powder XRD pattern (ICDD: 00-021-1276—rutile TiO_2_; ICDD: 00-021-1272—anatase TiO_2_; ICDD: 00-004-0850—fcc Ni); (**b**) M-H curve measured at 5 (red curve) and 300 K (blue curve); (**c**) ZFC (zero field cooling, blue trace) and FC (field cooling, red trace) curve plotted against the temperature; (**d**) BFTEM image; (**e**) HR STEM-HAADF image; (**f**) HRTEM image; and (**g**) respective FFT pattern of the highlighted region in Figure 1f, (<110> zone axis) fcc phase of bulk Ni could be identified.

**Figure 2 nanomaterials-13-01474-f002:**
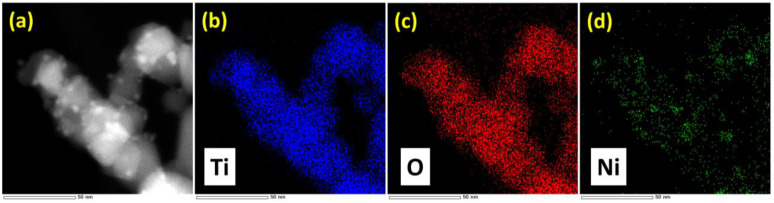
(**a**) STEM-HAADF image of 5Ni/TiO_2_ and EDS elemental mapping of the respective elements present in the sample; (**b**) Ti (blue); (**c**) O (red); and (**d**) Ni (green).

**Figure 3 nanomaterials-13-01474-f003:**
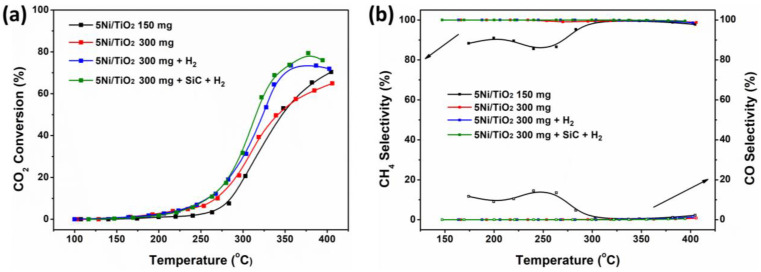
Catalytic activity under thermal heating: (**a**) % CO_2_ conversion and (**b**) % CH_4_ and % CO selectivity presented with respect to the reaction temperature for 150 mg of 5Ni/TiO_2_ (black trace), 300 mg of 5Ni/TiO_2_ (red trace), 300 mg 5Ni/TiO_2_ + H_2_ pre-treated (blue trace), and 300 mg 5Ni/TiO_2_ + SiC + H_2_ pre-treated (green trace) samples.

**Figure 4 nanomaterials-13-01474-f004:**
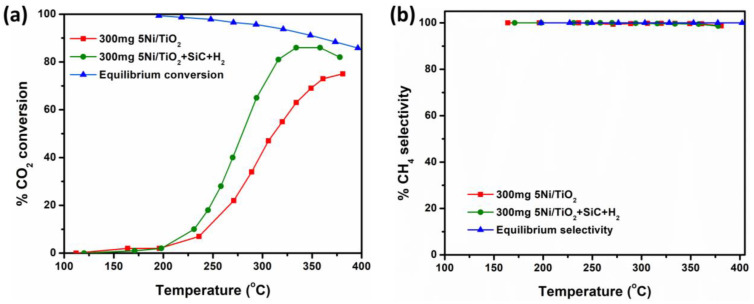
Catalytic activity under magnetically induced heating: (**a**) % CO_2_ conversion and (**b**) % CH_4_ selectivity of 300 mg of 5Ni/TiO_2_ (red trace) and 300 mg 5Ni/TiO_2_ + SiC + H_2_ pre-treated (green trace) samples. The blue trace corresponds to the calculated equilibrium CO_2_ conversion and CH_4_ selectivity at 1 bar pressure [36].

**Figure 5 nanomaterials-13-01474-f005:**
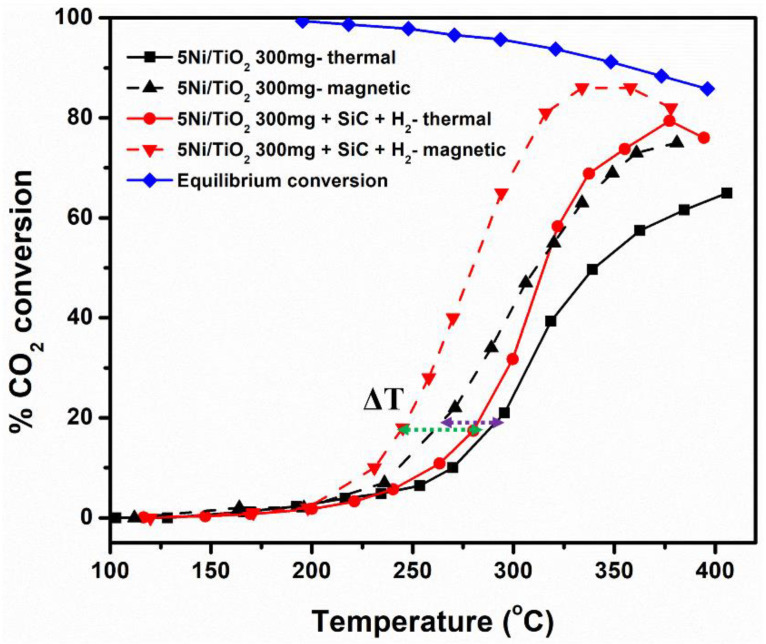
% CO_2_ conversion against temperature for 300 mg of 5Ni/TiO_2_ (black trace) and 300 mg 5Ni/TiO_2_ + SiC + H_2_ pre-treated (red trace); the dashed and the solid lines correspond to the reaction (GHSV = 20,000 mL·g_cat_^−1^·h^−1^; H_2_/CO_2_/Ar or N_2_ = 10/40/50; total flow rate 100 mL·min^−1^) carried out under magnetic induction and thermal heating conditions, respectively. The difference in temperature (ΔT) was shown as the dashed green (SiC diluted and H_2_ pre-treated bed) and violet arrows. The blue trace corresponds to the equilibrium CO_2_ conversion and CH_4_ selectivity [36].

**Figure 6 nanomaterials-13-01474-f006:**
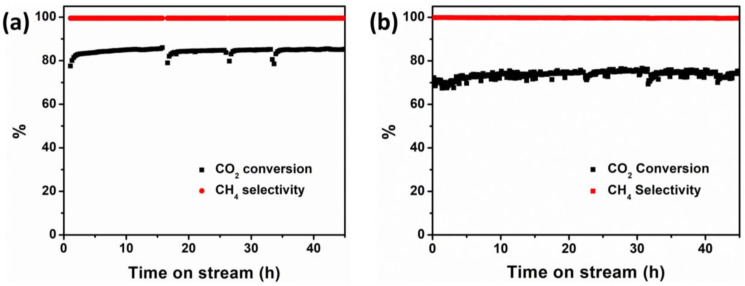
Activity of 300 mg 5Ni/TiO_2_ + SiC + H_2_ pre-treated catalyst; stability test of 45 h on stream at GHSV = 20,000 mL·g_cat_^−1^·h^−1^ (H_2_/CO_2_/Ar or N_2_ = 10/40/50; total flow rate 100 mL·min^−1^), (**a**) at 380 °C under classical heating, and (**b**) at 360 °C under magnetic induction heating. The dip in the catalytic activity under magnetic induction heating conditions arises due to the switching of the gases for security reasons.

**Table 1 nanomaterials-13-01474-t001:** Different reaction conditions used for thermal-heating-mediated methanation reaction.

Reaction Composition	Wt. of Catalyst (mg)	H_2_ Pre-Treatment	Use of SiC
5Ni/TiO_2_ 150 mg	150	No	No
5Ni/TiO_2_ 300 mg	300	No	No
5Ni/TiO_2_ 300 mg + H_2_	300	Yes	No
5Ni/TiO_2_ 300 mg + SiC + H_2_	300	Yes	Yes

**Table 2 nanomaterials-13-01474-t002:** CO_2_ conversion (%), CH_4_ yield (%), and CO yield (%) values for 5Ni/TiO_2_ tested under various reaction conditions.

Catalyst	% CO_2_ Conversion (Temperature/°C)	% CH_4_ Yield	% CO Yield
150 mg (thermal)	64.2 (380 °C)	63.4	0.78
300 mg (thermal)	60.5 (380 °C)	60.3	0.33
300 mg + H_2_·(thermal)	73.4 (380 °C)	72.8	0.54
300 mg + H_2_ + SiC (thermal)	77.9 (380 °C)	77.5	0.25
300 mg (magnetic)	63 (334 °C)	62.9	0.13
300 mg + H_2_ + SiC (magnetic)	86 (334 °C)	85.7	0.34

## Data Availability

Original data will be available upon request.

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
