# Peer review of "Catalytic Sabatier Process under Thermally and Magnetically Induced Heating: A Comparative Case Study for Titania-Supported Nickel Catalyst"

_nanomaterials, 2023, doi:10.3390/nano13091474_

Round 1

Reviewer 1 Report

The paper by S. Ghosh, B. Chaudret with coworkers is undoubtedly an important and actual research in the field of transforming CO2 into useful compounds. The research is seriously carried out, the manuscript is porperly organized and well written. It should be published after considering few notes:

1. Equations 1-3 need more explanations in regard of special designations (A, RF) for a reader who is not a specialist in the field

2. It should be explained how did the authors establish certain amounts of catalyst, the composition of the gas mixture and flow rate used in all experiments. Is it the result of the optimization of the reaction conditions? What impact could bring the change of the composition of the gas mixture? What is the role of the argon?  

Reviewer 2 Report

The author prepared a Ni supported on titania catalyst by an organometallic decomposition method, then CO2 methanation reaction is carried out under two different reaction conditions (thermal and magnetic induction heating conditions, respectively). The author concluded that the Fe wool and 5Ni/TiO2 composite bed is a stable and promising system for long-term methanation reaction under magnetic induction heating conditions. This study demonstrated the potential of magnetic heating mediated methanation. This article is recommended to be accepted after modification. There are some questions and suggestions:

1. In section 3.1, page 5, line 197, “The magnetic properties of the as prepared 5Ni/TiO2 sample are presented on figure 1. Figure 1b shows the magnetization vs field curve measured at 300 K and 5 K. The saturation magnetization values of the sample were normalized with respect to the total Ni conten……Herein, the possible paramagnetic contribution was removed”. Please explain in detail how performs normalization processing with respect to the total Ni content, and how to remove the possible paramagnetic contribution?

2. In section 3.2, page 10, line 386, “In contrast, the presence of dipolar interactions between the Ni nanoparticles was largely absent in the case of the spent catalyst obtained after the classical heating (figure S5b, ESI)”. How does the author determine the absence of dipole interactions between Ni nanoparticles based on the ZFC curve? Please explain in detail.

3. The author needs to supplement the XRD characterization of the catalyst after stability testing to compare the structural changes of the catalyst. And supplement CO2-TPD characterization.

4. Please unify the icon, font format, and ruler position details in each image (such as Figure 1 and 6).

5. Please add a blank experiment of only Fe wool without catalyst for CO2 methanation.

Round 2

Reviewer 2 Report

The authors have revise the manuscript following all previous suggestions, I agree for its publication in the present form.

Author Response

We appreciate that the reviewers acknowledged that our study is of interest to the audience of Nanomaterials. We thank the reviewer for recommending this manuscript for publication in Nanomaterials.